# Neonate Bloodstream Infections in Organization for Economic Cooperation and Development Countries: An Update on Epidemiology and Prevention

**DOI:** 10.3390/jcm8101750

**Published:** 2019-10-21

**Authors:** Jadwiga Wójkowska-Mach, Agnieszka Chmielarczyk, Magdalena Strus, Ryszard Lauterbach, Piotr Heczko

**Affiliations:** 1Department of Microbiology, Faculty of Medicine, Jagiellonian University Medical College, 31-121 Krakow, Poland; agnieszka.chmielarczyk@uj.edu.pl (A.C.); mbstrus@cyf-kr.edu.pl (M.S.); mbheczko@cyf-kr.edu.pl (P.H.); 2Neonatology Clinic, University Hospital, Jagiellonian University Medical College, 31-121 Kraków, Poland; ryszard@lauterbach.pl

**Keywords:** neonatal sepsis, bloodstream infections, very low birth weight neonates, epidemiology, prevention, probiotics

## Abstract

The term neonatal sepsis is used to describe a generalized bloodstream infection of bacterial, viral, or fungal origin which is associated with hemodynamic changes and other clinical symptoms and signs, however, there is no unified definition. There are no basic criteria regarding differentiation of early-onset sepsis (EOS) versus late-onset sepsis (LOS). Stratification used in studies on neonatal sepsis also rarely includes the general condition of the newborn according to unambiguous assessment at birth, which hampers the establishment of a clear, uniform epidemiological description of neonatal sepsis. We aim to review the published data about the epidemiology and microbiology of sepsis in Organization for Economic Cooperation and Development (OECD) countries. Data was also collected on sepsis prevention programs that can be implemented in neonatal units. The outcomes of interest were incidence or incidence density of EOS and LOS, microbiology of EOS and LOS, and data on the methodology of the research, in particular the criteria for inclusion and exclusion of newborns from the study. Pubmed, EMBASE, LILACS Embase, Scopus, and Google Scholar were used. For the preselection step, inclusion criteria included: “bloodstream infection” or “neonatal sepsis” (MesH), “very low birth weight”, and “country” full-text studies, human, and English language. Exclusion criteria included: studies published in languages other than English and studies available only as an abstracts. For proper selection, inclusion criteria included: information about epidemiology or microbiology bloodstream infection (BSI), study population and case definitions, exclusion criteria, narrative reviews, commentaries, case studies, pilot studies, study protocols, pediatric studies, and only clinical data (without microbiology or epidemiology) or studies with only one etiological factor analysis. The data review indicated the lack of an unequivocal, unified definition and no unambiguous basic criteria with regard to differentiation of EOS versus LOS. Among infants <1500 g, studies reported an EOS rate from 7% to 2%. For studies using other definitions (mostly all inborn babies), the rate of EOS ranged from 1% to 3%. The LOS incidences were much more varied among countries; the highest rates were in the multicenter studies focused on very low birth weight (VLBW) infants. The main pathogens in EOS are GBS and Gram-negative bacteria in LOS. Our review data shows that LOS microbiology is very diverse and that Gram-positive cocci, especially staphylococci, predominate versus Gram-negative rods. Unfortunately, the lack of uniform, international prevention programs results in high newborn morbidity and insufficient postnatal prevention of late-onset infections.

## 1. Introduction 

Infection in newborns, especially mild cases, remains a significant problem of contemporary medicine. In highly developed countries, one reason for this situation is the increasing number of multiple pregnancies and preterm births [1], such as in the United States (US), as compared with the period 1990 to 2007; the proportion of deliveries before 28 weeks of gestation in 1990 was 71% and 77% in 2007, and pregnancies with a higher number of fetuses (three and more) were observed twice more often than previously [2]. The second reason for the increasing proportion of infants among hospitalized patients is progression of survival rate of newborns with very low birth weight (VLBW) in modern neonatal intensive care units (NICU), however, this situation is invariably associated with high incidence of both early and late infections [3,4,5,6,7,8].

The most frequently observed infection in neonatal wards is bloodstream infection (BSI). According to a European point prevalence survey (2011–2012, based on European Center for Disease Prevention and Control (ECDC) methodology), BSI accounts for 44.6% of all healthcare-associated infections (HAIs) [9]. 

Thus, we aimed here to review and summarize the recently published data that most closely describe the incidence and microbiology of BSI in high-income countries. We chose Organization for Economic Cooperation and Development (OECD) countries as these have comparable rates of survival among premature infants. Data were also collected on BSI prevention programs and procedures that can be implemented in neonatal units. 

## 2. Methods

### 2.1. Data Sources 

The outcomes of interest were incidence or incidence density of EOS and LOS, microbiology of EOS and LOS, and data on the methodology of the research, in particular the criteria for inclusion and exclusion of newborns from the study. In the epidemiological and microbiological part of the study it was a two-stage search, performed by two review authors using MEDLINE via Pubmed, EMBASE, LILACS Embase, Scopus, and Google Scholar from 1980 to September 31, 2018. Two review authors independently performed a search concerning epidemiology and microbiology of BSI. The search was restricted to OECD countries [10]. The literature review on BSI prevention and control was performed by three review authors. 

### 2.2. Search Strategy and Study Selection

For the preselection step, inclusion criteria included: “bloodstream infection” or “neonatal sepsis” (MesH), “very low birth weight”, and “country” full-text studies, human, English language. Exclusion criteria included: studies published in languages other than English and studies available only as an abstracts. 

For proper selection, inclusion criteria included: information about epidemiology or microbiology BSI, study population and case definitions, especially, data on BSI or neonatal sepsis microbiology and incidence rates with reported denominators, specified dates of the data collection period, and a description of the methods used including study population, the definition of BSI, and eligibility rules for early- and late-onset cases of BSI. Exclusion criteria included: narrative reviews, commentaries, case studies, pilot studies, study protocols, pediatric studies, and only clinical data (without microbiology or epidemiology) or studies with only one etiological factor analysis. 

Initially, other sources of the data on incidence rates of neonatal sepsis etc. in non-English journals and governmental reports were also reviewed, but methodology and definitions used in these sources did not appear to be adequately precise, and therefore only the search described above was then presented. 

For proper selection, two author reviewers independently reviewed the titles and all abstracts. A study was eligible for inclusion if the epidemiology of BSI was provided at the national, regional, or multicenter level. In the case of a dearth of multicenter data, single-center study data were also included. 

From the included studies, the following information data were extracted: country, case definition, study population, as well as and inclusion and exclusion criteria, time period, incidence rates and microbiology; if possible, we also extracted birth-weight specific BSI incidence rates and specific microbiology according to early-onset (EOS) or late-onset cases of BSI (LOS) or neonatal sepsis (Table 1). When there were multiple publications from the same country, the data from the most recent publication were only included once.

## 3. Results 

A total of 4871 articles were identified, 199 articles were included in the proper selection stage after being screened according to titles and abstracts. Finally, 35 studies which fulfilled the eligibility criteria were included for the review (Appendix A).

The included studies represented data from 27 of the 36 OECD countries, primarily across Europe, Asia, and the US (Table 1). There were more available data on incidence rate (30 studies, 26 countries) than on microbiology of early-onset EO and late-onset LO BSI (28 studies, 25 countries). The country, time period, case definition applied, and incidence rates of BSI in the included studies are shown in Table 1. 

Fourteen studies were multicenter studies [11,12,13,24,26,27,30,33,36,37,38,39,40,42,44,45] one was a regional study [20], and five were national studies [22,23,24,27,34,35,41], however, twelve studies were single-center studies [14,15,16,17,18,19,25,28,29,31,32,43] (Table 1). Most study populations comprised inborn babies admitted to the neonatal unit. Inclusion criteria were based on birth weight (mostly below 1500 g) or gestational age, most accounted for all inborn babies (17 studies). In two studies, from Italy and Netherlands, the surveillance included only neonates with catheters (peripheral or central venous catheter, CVC). One study included a national cohort of extremely premature infants, Norway. With the exception of studies from the US (1991–1993), Austria (1988–1994), and France (1984) all studies included data from the last 20 years. The studies from Belgium, Finland, Australia, and New Zealand, and Israel covered the longest periods. 

Limited data were retrieved from some regions and many countries, but 25.7% of the OECD countries were not represented. Furthermore 12 (35.3%) studies were single-center studies with unknown representativeness.

### 3.1. Case Definitions for BSI 

It is generally accepted to use the term neonatal sepsis to describe a systemic condition initially evoked by bacteria, viruses, or fungi and which is accompanied with hemodynamic changes and other clinical manifestations leading to substantial morbidity and mortality in neonates, however, despite many years of clinical experience in diagnosis and treatment of neonates with confirmed or suspected sepsis, there is no commonly accepted definition of neonatal sepsis. Sometimes, the definition of sepsis erroneously includes isolation of a pathogen from normally sterile body fluid, such as blood or cerebrospinal fluid, which laboratory finding should be described as bacteremia. Since practically all clinical features and laboratory data of sepsis are induced by a burst of the potent proinflammatory cytokines, the term clinical sepsis (or systemic inflammatory response syndrome, SIRS) has also been used in literature in parallel to neonatal sepsis [46,47]. From an epidemiological point of view sepsis is a bloodstream infection (BSI).

Unfortunately, our data review indicated the lack of an unequivocal, unified definition. There are no unambiguous basic criteria with regard to differentiation of EOS versus LOS. Stratification in the research also rarely includes newborns’ general condition according to unambiguous assessment at birth; most program data refer to “inborn babies” without additional qualifying criteria. Both of these factors make it impossible to establish a clear description of the epidemiology of neonatal sepsis. 

In multicenter studies targeting VLBW neonates, the most commonly applied definition was that of the Centers for Disease Control and Prevention (CDC), used in three studies. The definition used in the Neo-KISS (a surveillance system for VLBW infants in Germany) and ECDC was applied in two studies (Poland and Germany) (Table 1). The remaining studies used their own definition, based on a combination of clinical signs and different time to recognition of EOS and LOS, mostly 48 or 72 h of life, but sometimes this was five or seven days (Table 1). 

### 3.2. Comparison of BSI Rate by Case Definition, Date of Onset (Early and Late), and Birth Weight Category

We present the rate of BSI by case definition, gestational age, and birth weight category (Table 2). Among infants <1500 g, studies using CDC or Neo-KISS definitions reported an EOS rate of 7% in Poland, 5% in Spain, and about 2% in the US and Israel. For studies using other definitions (mostly all inborn babies), the rate of EOS ranged around 1% to 3%, except 6.6% in Denmark. In a Norwegian national cohort of extremely premature infants, the EOS rate was 3.6% (Table 2). 

The LOS incidence rate was much more varied among countries; the highest rates were in the US, South American countries, Poland, and Spain in multicenter studies focused on VLBW infants. In a few studies, incidence rates were stratified by birth weight or gestational age (Table 3), the incidence rates were surprisingly similar in the US and Poland, and rates in Germany were lower in each newborn group (Table 3). 

In studies including or limited to catheter-associated BSI, the central line-associated bloodstream infection (CLABSI) incidence was comparable. In the Netherlands, this was 18.1/1000 patient days (pds) and 8.6/1000 pds in Poland. In Italy, this incidence was 4.7/1000 to 11.6/1000 pds, depending on birth weight, similar to an Australian study reporting 1.2/1000 to 3.5/1000 CVC-pds. Completely different results were found in the analysis of peripheral venous catheter -associated bloodstream infection (PLABSI) incidence. In Australia, this was 0.67/1000 peripheral line-pds, but the incidence was much higher in Poland, with 10.5/1000 peripheral line-pds.

There were some differences in the epidemiology of neonatal infections (both EOS and LOS) among the analyzed studies included in this review. The main source of these differences were most likely distinctions used by the authors in applying various parameters to describe the status of neonates at birth or the risk of infection, as well as time lapses between birth and the onset of infection, such as 48 h of life or 72 h of life or seven days. Likewise, different groups of neonates were used as denominators, i.e., all infants born versus VLBW newborns only. Moreover, different definitions of sepsis were applied. In a few studies, the basis for the diagnosis sepsis was a single positive blood culture and prolonged antibiotic therapy. No validation of the detection and qualification of sepsis cases was found in any studies.

The most important factor that was also closely related to intensive care in neonatal wards seemed to be stratification of cases into groups according to the risk of infection. Unfortunately, this methodology was applied in only a few studies, however, the most frequently used qualification was based on gestational weight as this parameter is easy to obtain, independent from other variables, and involves no danger of underestimation in comparison with gestational age, which is more subjective because it relies on cooperation between the obstetrician and parturient. Stratification of neonates according to gestational weight should be preferred in studies on the epidemiology of neonatal infections as it is used in two of the most important research projects on neonatal infections, the National Healthcare Safety Network (NHSN) and ECDC programs. 

### 3.3. Microbiology of BSI

In EOS, Gram-positive bacteria predominated, particularly group B *Streptococcus* (GBS); the highest percentage values were reported in French (58.5%) and British (43%) studies (Table 4). In Swedish research, a similar number of infections were caused by GBS (20%), coagulase-negative staphylococci (CoNS, 30%) and *Escherichia coli* (25%). The situation differed in South Korea and Denmark, where the main pathogen in EOS was *S. aureus* (48% and 36.6%). Mexico and Turkey dominated CoNS with a prevalence of 55.5% and 60.9%, respectively (Table 4). 

Among Gram-negative bacilli, the most frequently occurring was *E. coli*, which was associated with approximately 20% of cases. *E. coli* predominated in EOS in a study from Israel and two studies from North America; with respect to the latter, *E. coli* occurred in 33.4% of cases, whereas in the US alone, the rate was 44%. In early infections in Poland, GBS was common (20%), but an additional problem in that country is infections caused by *Klebsiella pneumoniae* (22%), also in EOS (Table 4). 

Infections caused by fungi in EOS accounted for 2% to 3% of cases and the highest level was 8% in Poland (Table 4). 

In LOS, CoNS was significantly predominant, from 24.2% in Australia to 75% in the UK and 85% in the Netherlands, most often in 30% to 50% of infections. France differed in that more than half of infections (55.5%) were caused by *E. coli.* In Japan, the main pathogen in LOS was *S. aureus* (26% and similar in Ireland, 26.9%) apart from *E.coli* and *Klebsiella* and other Gram-negative bacillis (24%); unexpectedly *Pseudomonas* spp. were also common (12%) (Table 5). In South Korea, among Gram-positive bacteria, both CoNS and *S. aureus* were predominant (37.5% and 36%, respectively). Infections caused by Gram-negative pathogens in LOS ranged from 7% in Finland to 64.4% in United Kingdom and yeast-like fungal infections were more common than in the case of EOS, from 2% in Switzerland to 18.8% in Turkey (Table 5). 

CLABSI were very strongly associated with CoNS; this was the case in Dutch (85%) and Australian (24.2%) studies, in line with the researchers’ expectations, however, in an Italian study of CLABSI, half of infections (50%) were caused by yeast-like fungi (Table 5). 

In summary, a predominance of Gram-positive cocci, especially GBS, was confirmed among bacteria isolated from EOS cases. CoNS was most often found in LOS related to CLABSI and PLABSI, although a broader range of microorganisms, including Gram-negative rods, was noted in LOS cases (Table 5). 

Data on the microbiology of neonatal sepsis collected in this review confirm already known discrepancies between the etiologies of EOS and LOS, as pathogens isolated from EOS cases are mostly derived from the mother’s vaginal and skin microbiota. There are, however, notable exceptions, mostly related to GBS infections, as has been found and documented by molecular typing where very early infections are caused by horizontally transmitted hospital GBS strains [48]. Accordingly, prophylaxis for early-onset neonatal sepsis can be attributed to surveillance of antenatal infections performed under the framework of obstetric care by qualified personnel, although standard surveillance in neonatal wards should start immediately after birth to protect infants from early infections transmitted during or immediately following labor. 

It should be noted that this review of infections reported and analyzed in high-income countries revealed that most infections (both EOS and LOS) are caused by GBS, *S. aureus,* and *E. coli*, regardless of the geographical and cultural differences among patients. This may also suggest that the epidemiology of neonatal infections in these countries reflects high levels of standardization of intensive care. The unusual prevalence of BSI *Candida* spp. in Italian neonatal intensive care units NICUs may be an early reflection of a new shift in neonatal sepsis etiology to emerging infections caused by *C. auris*. Owing to difficulties in identification, *C. auris* isolates may easily be misdiagnosed as other *Candida* species. Infections caused by this new pathogen have been increasingly reported from India, China, and also the US [49], although the US data included in this review do not indicate this tendency. 

### 3.4. Prevention and Control of BSI

Actually, only EOS can be prevented using a proven strategy based on appropriate administration of antibiotics (maternal intrapartum antibiotic prophylaxis, IAP) to stop vaginal colonization by GBS and prevent bacteria from reaching the newborn’s oral cavity and upper respiratory tract. At the moment, there are no other approved and implemented methods to prevent EOS and LOS in neonates. It should be mentioned that there are new strategies to detect the threat of EOS, regardless of its etiology, on the basis of serial physical examinations, however, this approach cannot be regarded as preventive. These strategies are implemented in NICUs to reduce laboratory costs and to limit overuse of antibiotics [50,51]. The most commonly used among these strategies are related to antibiotic stewardship principles adopted in many NICUs, sometimes under pressure from global campaigns for the prudent use of antibiotics promoted by international and national organizations such as the World Health Organization (WHO), the Infectious Diseases Society of America, the CDC, and the ECDC. In other cases, these strategies are adopted in NICUs in a more reasoned way, based on studies showing the effectiveness of such restrictions. Indeed, there are several studies indicating that monitoring of antibiotic prescribing can result in reduction of unnecessary antibiotic use in the NICU [52]. 

### 3.5. Intrapartum Antibiotic Prophylaxis (IAP) 

The incidence of GBS-caused EOS before introduction of IAP into practice was three to four per 1000 live births. When the first CDC guidelines for IAP were issued in the US during the 1990s, this rate had declined to less than 0.25/1000 live births [52,53]. After the CDC, the American Academy of Pediatrics (AAP), the American College of Obstetricians, and other US organizations issued the most current recommendations which are to be followed for the administration of GBS intrapartum prophylaxis. This procedure is now accepted by the overwhelming majority of national guidelines. Recent reviews on the effectiveness of IAP show that most cases of EOS caused by GBS can be prevented. There are now two methods, routine culture-based screening and risk-based management, which are used to identify mothers requiring IAP during labor [54]. This policy has been implemented in over 90 countries, although microbiological screening for GBS carriage has only been used in 30% of the countries and clinical risk factor screening has been used for decision making in the remaining countries [55]. 

IAP has some limitations and links to adverse short- or long-term neonatal effects [56], although there is only one report on serious outcomes of IAP, such as cerebral palsy. Seven observational studies showing that IAP alters the infant microbiome were recently analyzed by Seedat [57], however, the clinical significance of the alterations is unknown. There are also observational reports on increased antimicrobial resistance of the gut microbiota bacteria after IAP, however, the data contained in related studies are rather not well documented. Another long-term negative effect of IAP on gut microbiota, is an increased risk of noncommunicable diseases, such as allergies or obesity, which has been observed in neonates exposed to antibiotics during early infancy [58]. Still, it is too early to estimate the negative impact of IAP and early antibiotic treatment of the neonates and their long-term effects. Therefore, it is possible that in the future, GBS vaccination will be applied as EOS prophylaxis during pregnancy either alone or in combination IAP. Most probably, conjugated polyvalent vaccines against both EAO and LOS causing serotypes of GBS will be used. Such practice may decrease GBS infection rates in neonates more effectively than IAP alone [59].

The combined prophylaxis may also prevent late-onset GBS infection, which accounts for approximately 40% of all GBS neonatal infections. Preterm infants are particularly susceptible to late-onset GBS infections. Actually, neither GBS IAP nor any other neonatal EOS prevention have an effect on late-onset bacterial infection.

### 3.6. Implementation of Risk Stratification Strategies for Empirical Use of Antibiotics in Preterm Neonates 

There are other activities that can be regarded as a form of prophylaxis in newborns with risk factors for EOS. Management is well described in official guidelines of the CDC [60], the AAP [61], and the National Institute for Health and Care Excellence [62]. The main component of management is empirical antibiotic therapy given to asymptomatic neonates with known risk factors for EOS, with special regard to chorioamnionitis. Such an approach may reduce sepsis-related morbidity [60], although it has been questioned by some authors [63] as it leads to treating with antibiotics infants without sings of infection. Such overuse of empirical antibiotics may pose a hazard to unaffected infants. Generally, exposure of neonates to antibiotics, either related to IAP or to antibiotic prophylaxis during cesarean delivery or to empirical antibiotics given postnatally to preterm babies, is common. There is growing evidence that exposure to antibiotics is related to increased mortality in premature infants [64]. Moreover, prolonged application of antibiotics in preterm neonates, even without proven infection, may increase the risk of necrotizing enterocolitis (NEC), bronchopulmonary dysplasia, retinopathy, and brain damage [65,66]. 

Studies on colonization of the gut in newborns and establishment of the gut microbiota show that antibiotics given after birth alter the microbiota composition, enabling its early colonization with potential pathogens derived from the hospital environment and inhibiting colonization with bifidobacteria found in the gut microbiota of neonates born vaginally from healthy mothers [67]. Prolonged, empirical antibiotic treatment severely impairs diversity of the gut microbiota, and promotes pathogens associated with neonatal sepsis which are only minor members of the healthy microbiota [68].

Both the untoward effects related to empirical antibiotic treatment of anyway well-appearing term or late-preterm newborn with risk factors for sepsis and the risk of less significant but common NICU complications, argues in favor of a more carefully thought out management of newborns with sepsis risks [51]. Therefore, to limit antibiotic overtreatment related to empiric therapy given to asymptomatic newborns, new management strategies based on scheduled serial clinical observations, online sepsis risk calculations, and optimized antibiotic dosages have been recently proposed as an alternative to unlimited empiric antibiotics and further evaluation [69,70].

### 3.7. Surveillance 

External, nonmaternal risk factors are associated with high intensity of various medical procedures related to the care of newborns with VLBW. These include the use of corticosteroids in newborns with impaired breathing that lower their own neonatal immune response [71] and the necessity for diagnostic and therapeutic invasive procedures (intravascular catheters, respiratory support, and parenteral nutrition), which are risk factors of neonatal LOS [72,73,74,75]. The risk of colonization with undesirable microorganisms also increases with long-term hospitalization and intensive care of a newborn, especially in overcrowded wards with insufficient staff who are overworked [76,77,78,79]. In the US, the average hospital stay for newborns with birth weight <800 g is 112 days, and a central catheter (including an umbilical catheter) and mechanical ventilation are used in this group of premature infants on average for 44% of hospital days in the NICUs ward in newborns with extremely low birth weight, and in up to 11% in newborns with birth weight >2500 g [80,81]. In Poland, the average stay of a newborn with VLBW with an infection until it reaches 1800 g is 44 days, and the CVC utilization ratio is 45% [82].

According to recent opinions, it is possible to prevent at least some of the health care-associated infections in the NICU with active surveillance. A successful program for the neonatal sepsis surveillance should be based on several activities which include: continuous collection of data on the incidence of the systemic infections; monitoring the spectrum and sensitivities of organisms grown from these infections, controlling utilization rates of the invasive devices; and evaluating interventions and practices implemented to improve the quality of care using bundles. The program should be discussed and constructed by an infection control team or committee with multilevel representation from the unit personnel, preferably separately from but in cooperation with the all-hospital committee.

It has been confirmed that the introduction of the bundles is associated with a reduction in late-onset systemic infections among infants in the NICU [83,84,85]. The bundles, constructed separately for individual procedures or periods of care, are written locally to better adapt them to the specificities of a unit. Bundles are usually procedures based on evidence-based best practices, which improve care on individual patient. The bundles when applied together, result in substantially greater improvement. For example, activities of the central line bundle, as described by Pharande, include strict hand hygiene, maximal sterile barrier precautions, proper skin antisepsis, aseptic technique, optimal site choice for line insertion, transparent sterile dressing, surveillance, and documentation [85]. According to the authors, implementation of the bundle during a 15-year period resulted in a constant decline in the incidence of LOS and CLABSI in their NICU, in spite of an increase in admissions to the NICU, an increase in central line utilization, and no decrease in the number of high-risk infants. Thus, implementation of multiple infection control bundle practices, together with well-coordinated team efforts and the growing engagement of staff, leads to reduction of the nosocomial infection rates and also strengthening the improvements for a longer time.

The positive results of the implementation of the central line insertion bundle in the NICUs should lead to future implementation of other preventive practices, based on experiences from adult ICUs, for example, a postinsertion care bundle which consists of daily inspection of the insertion site; site care if the dressing is wet, soiled, or has not been changed for seven days; documentation of an ongoing need for the catheter change; proper application of a chlorohexidine-impregnated sponge at the insertion site; scrupulous hand hygiene before supervision of the intravenous line; and disinfection of the infusion hub for 15 s with alcohol scrub before each entry. Such a CVC postinsertion care bundle was implemented in the adult ICU of the Denver Department of Veterans Affairs and its application was associated with a significant reduction of CLABSI in a setting in which compliance previously introduced central line insertion bundle was already high [84].

### 3.8. Probiotics 

Commensal human flora is an extremely valuable first line of defense of the organism, limiting the possibility of invading foreign microorganisms and activating its own immune system. The newborn is in the sterile environment of the mother’s womb without exposure to microbes until delivery, and colonization of the newborn in cases of a healthy pregnancy takes place in direct relationship with the mother and the home environment. However, newborns admitted to NICUs are subjected to exposure and colonization with bacterial strains originating from the birth or surgical wards, including from medical personnel. These microorganisms are often characterized by higher resistance to antibiotics and multidrug resistance mechanisms, or even higher virulence [86,87,88,89,90,91]. 

Meta-analyses have confirmed a reduction in the risk of NEC through the use of probiotics and also the risk of LOS [92,93] in preterm infants, although there is disagreement about the latter entity. Zhang et al. recently stated that probiotic supplementation is safe and may significantly reduce the incidence of LOS in preterm neonates in the NICU [93] but, conversely, meta-analysis of Olsen et al. on prophylactic use of probiotics in preterm infants, did not confirmed a significant reduction in sepsis rates, although a trend toward this effect was noted [94]. Both meta-analyses found a significant reduction in NEC rates, as well as, in general, mortality among neonates after probiotic supplementation but probably in exclusively human milk-fed preterm infants only [95]. 

The postulated mechanisms of the benefits of probiotics include tight junction (gut barrier) enhancement, modulation of the gut immune system such as toll-like receptor 4 (TLR4), nuclear factor-B, and proinflammatory cytokines, and direct inhibition of gut pathogens, however, the commonly accepted theory on mechanisms of LOS, which is mostly caused by CoNS, is that these skin microbiota members colonize the skin insertion site and catheter hub. From a contaminated hub, microorganisms migrate along the surface of the catheter and enter the bloodstream. This view is supported by in vitro studies, demonstrating that CoNS and, particularly *Staphylococcus epidermidis,* are able to adhere to plastic surfaces and build a biofilm on them. 

The mechanisms of LOS, described above, are evidently not related to the protective activity of probiotics, which is based on correction of the altered gut microbiota toward the elimination of pathogens and tighten gut barrier to prevent bacterial translocation from gut to bloodstream. There are, however, reports supporting the alternative mechanisms of LOS by demonstrating neonatal sepsis cases which have not been related to catheter use but to translocation of bacteria owing to the increased intestinal permeability typical in premature neonates [96,97,98]. Therefore, it is possible that at least some LOS cases are related to translocation of the potential pathogens colonizing the gut mucosa. Such a mechanism may explain a positive effect of properly selected probiotics on the gut microbiota and, consequently, on the prevention of LOS. We recently observed that oral treatment of VLBW neonates with a mixture of two probiotic bacteria, *Bifidobacterium breve* and *Lactobacillus rhamnosus*, was related to significantly lower rates of LOS caused by staphylococci and especially CoNS. These findings may support a view that one mechanism of LOS caused by CoNS may be based on translocation of CoNS, present in large numbers in the gut of VLBW infants, to the bloodstream and not only on colonization and biofilm building on catheter surfaces by CoNS derived from skin microbiota. This hypothesis should be verified in specially designed studies.

Since recent studies stress the importance of the intestinal microbiota alterations in premature neonates as an elementary cause of many diseases in childhood, it seems that a necessary first step is to design novel approaches that correct the microbiota using selected probiotic bacteria as a pioneering organism [99]. Such an approach may lead to early probiotic interventions to prevent LOS in high-risk infants. Well-characterized and clinically proven probiotics should be used for this purpose. 

Another concept related to probiotic use is the application of probiotics to pregnant women for the prevention of general preterm morbidity and mortality. Such a possibility has been tested and evaluated.

One of the most exciting scientific advances in recent years has been the realization that commensal microorganisms (our microbiome) play vital roles in human physiology with respect to nutrition, vitamin synthesis, drug metabolism, protection against infection, and recovery from illness. Recent data show that loss of “health-promoting” microbes and overgrowth of pathogenic bacteria (dysbiosis) among patients in the ICU appears to contribute to nosocomial infections, sepsis, and poor outcomes [100]. Overall, accumulating data regarding probiotic and synbiotic therapy reveal a need for definitive clinical trials of these therapies, as recently performed in healthy neonates. Future studies should target administration of probiotics and synbiotics with known mechanistic benefits to improve patient outcomes. Optimally, future probiotic and synbiotic studies will be conducted using microbiome signatures to characterize actual ICU dysbiosis and determine, perhaps even personalize, ideal probiotic and synbiotic therapies. 

Twelve clinical trials have already been analyzed [101] with the authors’ conclusion that although safe, it is too early to state whether this preventive strategy is beneficial owing to inconsistency and imprecision of the data. Except for the use of probiotics and prebiotics, prophylactic use of different drugs which may help to promote a healthy gut microbiota and maturation of the immune system in preterm infants is recently proposed. Among these, use of lactoferrin as a promising dietary supplement had been considered, however, the effectiveness of lactoferrin to prevent LOS and NEC in preterm infants and its safety was regarded as controversial. It is noteworthy to mention, that the latest meta-analysis confirms that lactoferrin could significantly reduce the incidence of NEC and LOS and decrease infection-related mortality in premature neonates without obvious adverse effects [102,103].

## 4. Discussion 

Surveillance of infections in many different patient populations is well described and functioning, starting from uniform, supranational definitions. Unfortunately, for patients in NICU wards, there is no consensus regarding the principles of surveillance. These differences make it impossible to conduct an unequivocal assessment of the existing epidemiological and microbiological situation in various health care systems, as well as the possibility of implementing elements of prevention. Lack of uniform research tools precludes the design and implementation of supralocal research programs concerning implementation of the various possible solutions described above (Table 6). 

The most prospective and natural way to prevent sepsis, in neonates at risk in the future, seems to be the correction of the gut microbiota composition during the very short but critical period of life just after birth, to enable establishment of bacteria derived from the vaginal microbiota and prevent gut colonization with bacteria from the hospital environment. There are multiple studies showing that preterm infants born by cesarean delivery, who are very prone to sepsis, have altered or disrupted gut microbiota containing potentially pathogenic gram-positive and gram-negative organisms that are often resistant to multiple antibiotics [67]. 

Although the molecular mechanisms of the early priming immune system period in humans have not been defined, thanks to new animal experiments, there are new data showing that mechanisms of acquiring the gut microbiota in infancy depend on interactions between bacterial, as well as host factors. It was found that the first microbes introduced into young, germ-free, genetically identical mice exert the strongest influence on the gut microbiome [102,103]. 

Recently, a new mechanism has been demonstrated, which regulates bacterial colonization that is active only during the early neonatal period but also influences life-long microbiota composition. Fulde et al. showed that expression of the flagellin receptor TLR5 in the gut epithelium of mouse neonates is age-dependent [110]. They demonstrated, using microbiota transfer experiments in neonate, adult wild-type and TLR5-deficientgerm-free mice, that epithelial TLR5-mediated REG3γ production is critical for the counter-selection of the mucosa-colonizing flagellated bacteria and its expression occurs in neonatal period. This discovery might explain why environmental factors that disturb establishment of the normal microbiota in early life period can affect immune homeostasis and health in adulthood. 

There is also new evidence suggesting that gut microbiota composition can have a meaningful impact on the composition of the serum immunoglobulin A (IgA) repertoire and bone marrow plasma cell pool, which may build protection against systemic infection. Wilmore et al. supplemented the gut microbiota of conventional mice with selected proteobacteria and demonstrated that the modification of the microbiota composition caused elevation of the serum IgA concentrations which were correlated with colonization of the bone marrow by large numbers of IgA-secreting plasma cells and marked changes in the serum IgA pool [105].

The actual state of knowledge on gut colonization implicates that the gut microbiota composition is shaped during early infancy but its effect on health is prolonged over adulthood and provides protection against systemic infections. Thus, proper timing and proper bacteria are the crucial factors that may determine successful artificial colonization of neonates at risk. This approach will become more successful when these and other factors are identified in humans and the effects of artificial colonization are proven in clinical trials. 

Although the antibiotic prophylaxis against early onset of GBS infection is effective, antibiotics are useless in preventing late onset of the disease in neonates. The introduction of a vaccine for pregnant women in the third trimester is likely to further reduce the burden of disease and provide benefits beyond the prevention of both EOS and LOS, including prevention of meningitis and disability following late-onset infections [59,105]. This makes development of the GBS vaccine an important and effective approach for prophylaxis. Up to now, different multivalent conjugate vaccines containing GBS polysaccharide antigens representing the main capsular serotypes of the bacteria have been developed. One of them, a trivalent GBS vaccine, was already successfully evaluated in a phase 1b/2 trial [106].

An immunogenic and protective mucosal vaccine based on inactivated GBS can be as effective as traditional ones but administered orally [107]. Gupalova recently reported the development of a live mucosal vaccine based on a probiotic strain, which is able to induce the appearance of pathogen-specific antibodies owing to the inclusion of antigen of the bacterial pathogen in the structure of the pili protein gene [108]. For this purpose, BAC protein DNA of GBS was integrated into the gene coding for the fimbrial protein D2 of *Enterococcus faecium* L3 [108], however, this approach seems to have some disadvantages, for example, a risk of using *E. faecium*, a bacterial species that is an important human pathogen, is the acquisition of extensive antibiotic resistance [111]. 

*E. coli* is the second most common organism associated with EOS and other serious bacterial infections in neonates. It is postulated that the common use of IAP has changed the incidence of GBS and *E. coli* as causes of EOS and LOS. Moreover, increased use of IAP might promote the emergence of multidrug resistant microorganisms that cause EOS, as well as LOS. The most likely way to prevent it could be an anti-*E. coli* vaccine, that should be effective against strains that are associated with major diseases and resistant to multiple drugs [109,112]. This solution is particularly desirable for populations of special interest, such as pregnant women at risk of preterm labor.

## 5. Conclusions

In the present review, we revealed a diverse pattern in the epidemiology and etiology of neonatal sepsis among 36 OECD countries located on different continents, although the basic data are similar and remain quite homogenous. This may indicate that it is possible to discuss the potential construction and implementation of common and effective surveillance programs for infection prevention and control (IPC) in neonatal wards. 

According to the WHO recommendations [1], surveillance of HAI, including neonatal sepsis (or BSI) is critical to inform and guide IPC strategies. Such surveillance should use standardized definitions and methodology not only for continuous, everyday infection surveillance but also as a method for evaluating the quality of the data.

Further studies and analyses are certainly needed to achieve modern surveillance programs that will take into account different requirements and various capabilities in the implementation of such programs. The most important obstacle does not seem to be access to modern medical technologies but rather the level of knowledge and skills among professional infection control workers, including interpretation of the microbiological data. 

The most prominent differences in the epidemiology of neonatal sepsis collected and presented in this review were observed not in epidemiological indicators but rather in the microbiology of infections, especially LOS. Without in-depth studies, it is difficult to find a cause for these discrepancies, but one possible factor may be differences in the quality of microbiology and laboratory capacities, which are essential for reliable HAI surveillance. The microbiological data on sepsis etiology and pathogen resistance patterns, especially according to EOS and LOS, also provide relevant information on policies of antimicrobial prophylaxis or therapy and antimicrobial resistance-related strategies and interventions. 

New approaches to IPC in NICUs underline the necessity to apply multimodal strategies that take into account cooperation among different groups of health care personnel. Thus, a uniform training program in infection control in neonatology is urgently required, together with surveillance programs. Such programs should also involve modern microbiological approaches based on both classical and molecular methods of detection, characterization, and epidemiological typing. 

### Study Limitations

This review is based on full texts (original articles or concise communications) written in English. We did our best to find useful data published in other languages, as suggested by Higgins et al. [113], however, not all of the figures we were looking for can be easily found. We were unable to find some useful information on the methodology of infection control in the non-English sources, and therefore we decided to limit the reviews to those written in the English language. 

## Figures and Tables

**Table 1 jcm-08-01750-t001:** Characteristics of the collected studies on neonatal sepsis and bloodstream infections incidence rate in Organization for Economic Cooperation and Development (OECD) countries (in alphabetical order).

Country	Population: Criteria of Including	National/Regional Network or Database	Case Definition	References
Austraila & N. Zealand, 2002–2012	all inborn babies	Australasian Study Group for Neonatal Infections (ASGNI)	EOS: 48 h of life	[11]
Australia, 2008–2016	CLABSI and PLABSI in neonatal ICUs	Victorian Healthcare Associated Infection Surveillance System	CDC definitions, NHSN methodology, LOS: 48 h of life	[12]
Australia & N. Zeland, 2005–2007	all inborn babies, born <32 weeks of gestation	29 tertiary NICUs	EOS: 48 h of life	[13]
Austria, 1988–1994	all babies	1 center study	EOS: <7 days of life	[14]
Belgium, 2002–2011	all infants, inborn or hospitalised for at least 3 days	1 center study: tertiaty NICU	LOS: more than 72 h of life, only laboratory-confirmed BSI	[15]
Canada, 2005–2007	all inborn babies, born <32 weeks of gestation	26 tertiary NICUs	EOS: 72 h of life	[13]
Denmark, 2010–2013	all infants, inborn or hospitalised born <28 weeks of gestation	1 center study: tertiaty NICU	EOS: <7 days of life, only laboratory-confirmed BSI	[16]
Denmark, 2005	all infants, inborn or hospitalised	1 center study: tertiaty NICU	LOS: more than 48 h of life, only laboratory-confirmed BSI	[17]
Estonia, 2004–2008	all inborn babies	1 center study: NICU and paediatric intensive care unit	CDC definitions, only laboratory-confirmed BSI	[18]
Finland, 1999–2006	all inborn babies	1 center study	CDC definitions, only laboratory-confirmed BSI	[19]
France, 2004–2005	all inborn babies	Alsace, 9 tertiary NICUs	EOS: 72 h of life	[20]
France, 2007	all inborn babies	Alsace, 9 tertiary NICUs	LOS: between 72 h and 90 days of life	[21]
Germany, 2000–2005	VLBW infants	Neo-KISS	Neo-KISS definition, LOS: more than 72 h of life	[22]
Germany, 2010--2011	VLBW infants	Neo-KISS	Neo-KISS definition, LOS: more than 72 h of life	[23]
Greek, 2012–2015	neonates admitted to participating NNUs	16 NICUs participating in the neonIN infection surveillance network	EOS: 48 h of life, positive blood, cerebrospinal fluid or urine culture and were treated with at least 5 days of antibiotics	[24]
Ireland, 2001–2014	all newborn infants, only “culture-proven”	1 center study: tertiaty NICU	EOS: <7 days of life, LOS more than 7 days of life only laboratory-confirmed BSI	[25]
Israel, 1995–1998	VLBW infants survived at least 3 days	national cohort	LOS: more than 72 h of life	[26]
Israel, 1995–2005	VLBW infants	Israel Neonatal Network	undefined	[27]
Israel, 2007–2013	population-based study	1 center study	EOS: <7 days of life	[28]
Italy, 2006–2010	all inborn babies	1 center study	CLABSI only	[29]
Japan, 2006 to 2008	all newborn infants, only “culture-proven”	5 NICUs	only laboratory-confirmed BSI	[30]
Mexico, 2004–2007	neonates admitted to the neonatal ICUs	1 center study	LOS: more than 72 h of life	[31]
Netherlands, 2007	all infants with catheter	1 center study	CLABSI only	[32]
North America, 1997–2010	VLBW infants	313 NICUs	LOS: more than 72 h of life, only laboratory-confirmed BSI	[33]
Norwey, 1999–2000	all infants with gestational age of <28 weeks or birth weight of <1000 g	national cohort of extremely premature infants	EOS: diagnosed to day 7 of life	[34,35]
Poland, 2009	VLBW infants	Polish Neonatology Surveillance Network	Neo-KISS definition, EOS: 72 h of life	[36,37]
South America countries, 2001–2013	inborn VLBW infants	26 tertiary NICUs	LOS: more than 72 h of life	[38]
South Korea, 1997–1999	all infants, inborn or hospitalised	4 neonatal centers	LOS: more than 72 h of life	[39]
Spain, 2002–2005	VLBW infants	SEN1500 network by Spanish Society of Neonatology	undefined	[40]
Sweden, 2004–2007	extremely premature infants, born before 27 weeks of gestation and survived their first year of life	nationwide Swedish prospective cohort study	only clinical sepsis with negative blood culture and treated for a min. of 5 days	[41]
Switzerland, 2011–2015	all newborn infants admitted to tertiary care ICUs	the Swiss Pediatric Sepsis Study	EOS: 72 h of life and/or LOS if onset of infection was ≤48 h after admission, only laboratory-confirmed BSI	[42]
Turkey, 2003–2010	preterm infants with gestational age of <37 weeks, only “culture-proven”	1 center study	EOS: 72 h of life, LOS more than 72 h of life, only laboratory-confirmed BSI	[43]
United Kingdom, 2005–2014	undefined	neonIN: 30 NICUs	only laboratory-confirmed BSI treated with at least 5 days of appropriate antibiotics	[44]
USA, 1998–2000	VLBW and extremely premature infants	national: NICHD Neonatal Research Network	EOS: 72 h of life	[45]

CLABSI, central line-associated bloodstream infection; PLABSI, peripheral venous catheter-associated bloodstream infection; EOS, early-onset sepsis; CDC, Centers for Disease Control; LOS, late-onset sepsis; VLBW, very low birth weight; BSI, bloodstream infection; NICU, neonatal intensive care units,

**Table 2 jcm-08-01750-t002:** Comparison of the early-onset (EO) versus late-onset sepsis (LO)/bloodstream infections incidence rate (in alphabetical order).

Country	Rate of EO Sepsis	Rate of LO Sepsis	References
Australia & N. Zealand, 2005–2007	1.7%	15.1%	[13]
Australia, 2008–2016	n/a	CLABSI: 2.20 per 1000 CVC-days PLABSI: 0.67 per 1000 peripheral line-days	[12]
Austria, 1988–1994	1.9%	4.6%	[14]
Belgium, 2002–2011	n/a	7.1%	[15]
Canada, 2005–2007	1.3%	18.7%	[13]
Denmark, 2010–2013 / 2005	6.6%	7.6%	[16,17]
Estonia, 2004–2008	n/a	9.2% and 12.8 per 1000 pds	[18]
Finland, 1999–2006	n/a	3.2%	[19]
France, 2004–2005	1.08%	n/a	[20]
France, 2007	n/a	4.9%	[21]
Germany, 2010–2011	n/a	5.7 per 1000 pds	[23]
Greek, 2012–2015	0.4%	4.5%	[24]
Ireland, 2001–2014	0.1%	0.2%	[25]
Israel, 1995–1998	n/a	30.1%	[26]
Israel. 1995–2005	2.4%	n/a	[27]
Japan, 2006–2008	0.13%	0.61%	[30]
Mexico, 2004–2007	3.28%	1.08%	[31]
Netherlands, 2007	n/a	18.1 per 1000 pds CLABSI, catheter-associated BSI, only	[32]
North America, 19978–2010	1.0%	12.2%	[33]
Norway, 1999–2000	3.6%	9.7%	[34,35]
Poland, 2009	7.0%	25.3% CLABSI: 8.6 per 1000 CVC-days PLABSI: 10.5 per 1000 peripheral line-days	[36,37]
South America countries, 2001–2013	n/a	22.2%	[38]
South Korea, 1997–1999	29.3 per 1000 live births	7.2 per 1000 live births	[39]
Spain, 2002–2005	5.0%	29.4%	[40]
Sweden, 2004–2007	66%		[41]
Switzerland, 2011–2015	0.28 per 1000 live births	0.86 per 1000 live births	[42]
Turkey, 2003–2010	2.3%	12.9%	[43]
United Kingdom, 2005–2014	0.7 per 1000 live births and 5.5 per 1000 neonatal admissions	6.1 per 1000 live births and 48.8 per 1000 neonatal admissions	[44]
USA, 1998–2000	1.5%	21%	[45]

central line catheter-associated bloodstream infections, CLABSI; patient days, pds; peripheral venous catheter-associated bloodstream infections, PLABSI; CVC, central venous catheter.

**Table 3 jcm-08-01750-t003:** Comparison of the late-onset (LO) sepsis incidence rates stratified by birth weight or gestation age (in alphabetical order).

Country	Rate of LO Sepsis: Stratification by Birth Weight or Gestation Age				References
Australia and N. Zealand, 2005–2007	<25 weeks: 2.7	25–26 weeks: 2.8	27–28 weeks: 2.0	29–31 weeks: 2.2	[12]
Canada, 2005–2007	<25 weeks: 3.2	25–26 weeks: 2.3	27–28 weeks: 1.2	29–31 weeks: 0.7	[13]
Germany, 2010–2011	<1000 g: 8.5 per 1000 pds		1001–1499 g: 4.0 per 1000 pds		[22]
Italy, 2006–2010	<750 g: 11.6 per 1000 pds *	751–1000 g: 8.6 per 1000 pds *	1001–1500 g: 4.7 per 1000 pds *		[29]
Poland, 2009	<750 g: 44.6%	751–1000 g: 31%	1001–1500 g: 18.0%		[36,37]
USA, 1998–2000	<750 g: 42.8%	751–1000 g: 28%	1001–1500 g: 11.0%		[45]

* catheter-associated bloodstream infections, CLABSI only; pds, patient days.

**Table 4 jcm-08-01750-t004:** Share of the most common species of Gram-positive cocci, Gram-negative bacilli, and fungi in early-onset neonatal sepsis (in alphabetical order).

	Organism	Gram-Positive Cocci (%)				Gram-Negative Bacilli (%)			Fungi	References
Country		*Staphylococcus aureus*	Coagulase-negative staphylococci	Group B Streptococcus	others	Escherichia coli	Klebsiella spp.	others		
Australia and N. Zealand, 2002–2012		3.0	6.0	37.0	16.5	25.0	n/a	12.5	1.0	[11]
Australia, 2008–2016		Only LO sepsis								[12]
Austria, 1988–1994		6.3	12.5	28.1	21.9	0.0	9.4	18.8	3.1	[14]
Belgium		Only LO sepsis								[15]
Denmark		36.0	n/a	27.0	9.0	14.0	9.0	5.0	n/a	[16]
Estonia, 2004–2008		Only LO sepsis								[18]
Finland, 1999–2006		Only LO sepsis								[19]
France, 2004–2005		1.8	2.4	58.5	9.4	22.4	0.6	4.5	n/a	[20]
France, 2007		Only LO sepsis								[21]
Germany, 2002–2005		Only LO sepsis								[22]
Greek, 2012–2015		n/a	17.4	13.0	30.4	17.4	4.3	15.2	2.2	[24]
Ireland		12.6	14.1	38.5	n/a	14.1	n/a	n/a	n/a	[25]
Israel, 1995–1999		Only LO sepsis								[26]
Israel, 1995–2005		n/a	17.2	9.4	n/a	26.8	n/a	n/a	3.0	[27]
Italy, 2006–2010		Only LO sepsis								[29]
Japan, 2006 to 2008		Only LO sepsis								[30]
Mexico, 2004–2007		0.0	55.5	22.2	n/a	11.1	11.1	0.0	0.0	[31]
Netherlands, 2007		Only LO sepsis								[32]
Norway, 1999–2000		18.5	25.9	11.1	3.7	33.3	3.7	0.0	3.7	[34,35]
North America, 1997–2010		2.1	2.3	18.2	2.9	33.4	1.5	11.7	2.7	[33]
Poland, 2009		4.0	16.0	20.0	14.0	12.0	4.0	22.0	8.0	[36,37]
South America countries (including Chile), 2001–2013		Only LO sepsis								[38]
South Korea, 1997–1999		48.0	27.2	1.6	7.4	9.9	n/a	n/a	3.3	[39]
Sweden, 2004–2007		0.0	30.0	20.0	0.0	25.0	5.0	5.0	5.0	[41]
Switzerland, 2011–2015		2.0	8.0	38.0	23.0	23.0	n/a	6.0	0.0	[42]
Turkey		4.3	60.9	8.7	8.7	4.3	0.0	13.0	0.0	[43]
United Kingdom, 2005–2014		n/a	n/a	43.0	n/a	18.0	n/a	n/a	0.8	[44]
USA, 1998–2000		n/a	10.7	10.7	15.5	44.0	n/a	16.7	2.4	[45]

**Table 5 jcm-08-01750-t005:** Share of the most common species of Gram-positive cocci, Gram-negative bacilli, and fungi in late-onset neonatal sepsis (in alphabetical order).

	Organism	Gram-Positive Cocci (%)				Gram-Negative Bacilli (%)			Fungi	References
Country		Staphylococcus aureus	Coagulase-negative staphylococci	Group B Streptococcus	others	Escherichia coli	Klebsiella spp.	others		
Australia and N. Zealand, 2002–2012		Only EO sepsis								[11]
Australia *, 2008–2016		16.1	24.2	n/a	11.3	11.1	11.6	19.1 (including 8.8 Enterobacter)	5.6	[12]
Austria, 1988–1994		4.9	33.3	0.0	21.0 (Enterococcus spp. only)	1.2	2.5	22.2	18.5	[14]
Belgium		16.4	51.2	1.5	12.0	6.1	2.9	7.9	1.8	[15]
Denmark		5.8	26.9	n/a	n/a	1.9	n/a	3.8	1.9	[17]
Estonia, 2004–2008		5.5	48.6	n/a	8.3	n/a	7.3	23.8 (including *Serratia* spp. 12.0)	3.7	[18]
Finland, 1999–2006		7.0	65.0	n/a	6.3	3.0	3.0	1.0	9.0	[19]
France, 2004–2005		Only EO sepsis								[20]
France, 2007		12.7	13.6	7.3	n/a	55.5	n/a	n/a	n/a	[21]
Germany, 2000–2005		9.8	54.2	n/a	3.9	4.6	6.3	6.4	3.1	[22]
Greek, 2012–2015		0.5	31.5	0.2	7.0	13.0	19.4	17.7 (including 8.0 Enterobacter)	10.7	[24]
Ireland		26.9	22.1	7.7	11.1	10.6	10.6	n/a	n/a	[25]
Israel, 1995–1999		3.9	47.3	0.3	2.9	2.8	14.7	10.3	11.1	[26]
Israel, 1995–2005		Only EO sepsis								[27]
Italy **, 2006–2010		2.1	4.2	n/a	2.1	8.3	6.3	16.7	50.0	[29]
Japan, 2006 to 2008		26.0	12.0	7.0	14.0	12.0	5.0	24.0 (including 12.0 *Pseudomonas* spp.)	n/a	[30]
Mexico, 2004–2007		16.7	47.4	2.6	0.0	2.6	5.1	8.9	16.7	[31]
Netherlands **, 2007		2.5	85.0	n/a	2.5	2.5	2.5	2.5	2.5	[32]
Norwey, 1999–2000		12.0	47.0	9.0	2.0	1.0	8.0	10.0	10.0	[34,35]
North America, 1997–2010		15.4	28.3	3.1	6.8	6.2	6.8	9.6	10.5	[33]
Poland, 2009		7.8	62.7	n/a	6.6	6.6	6.8	5.9	3.8	[36,37]
South America countries (including Chile), 2001–2013		8.7	44.3	n/a	5.7	3.8	9.5	5.6	7.0	[38]
South Korea, 1997–1999		36.0	37.5	0.0	7.8	7.8	n/a	n/a	10.9	[39]
Sweden, 2004–2007		5.9	67.8	2.0	3.0	1.3	3.6	4.2	6.9	[41]
Switzerland, 2011–2015		15.3	36.5	15.3	9.1	24.7	n/a	13.0	2.3	[42]
Turkey		5.5	49.2	5.5	4.7	0.8	3.9	11.7	18.8	[43]
United Kingdom, 2005–2014		8.0	75.0	5.0	12.8	32.0	21.0	11.6	4.0	[44]
USA, 1998–2000		7.8	47.9	2.3	12.2	4.9	4.0	8.5	13.9	[45]

Notes: * central line catheter-associated bloodstream infections, CLABSI, and peripheral venous catheter-associated bloodstream infections, PLABSI; ** central line catheter-associated bloodstream infections, CLABSI only.

**Table 6 jcm-08-01750-t006:** Potential interventions in early- and late-onset neonatal sepsis prevention and control.

Implementation of Various BSI Prevention and Control Strategies	References
Decrease of GBS in EOS: ● maternal intrapartum antibiotic prophylaxis ● GBS maternal vaccination (future solution)	[50,51,52,53,54,55,104] [59,105,106,107,108]
Decrease of *E. coli* in EOS: ● *E. coli* maternal vaccination (future solution)	[109]
Decrease of EOS: ● risk stratification of asymptomatic neonates for empirical use of antibiotics	[60,61,62,69,70]
Decrease of LOS: ● surveillance with “bundle of care” strategy ● probiotics	[83,84,85] [92,93]

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
