# Peer review of "Neonate Bloodstream Infections in Organization for Economic Cooperation and Development Countries: An Update on Epidemiology and Prevention"

_jcm, 2019, doi:10.3390/jcm8101750_

Round 1

Reviewer 1 Report

The manuscript has been improved by the authors since the last version. However, there are several aspects that authors should review again:

1.- Abstract: please provide information on the methodology section.

2.- Methods:

2.1. Data sources and search strategy: Please provide a table with the number of records located in each database and the final records included in each database. In addition, it is very interesting to include the search date.

2.2. Selection of studies: The review included only the full text (original articles or concise communications) in English. The Cochrane handbook (2011) says "2.3.2: “So far as is possible, it is important to take an international perspective. The evidence collected should not be restricted by nationality or language without good reason, background information such as prevalence and morbidity should where possible take a global view, and some attempt should be made to put the results of the review in a broad context”. The authors should explain why they selected articles in English only. In addition, a short paragraph could be included in the limitations section.

Reference: Higgins JPT, Green S (eds). Cochrane Handbook for Systematic Reviews of Interventions Version 5.1.0 [updated March 2011]. Cochrane Collaboration, 2011. Available from www.cochrane-handbook.org

Author Response

1.- Abstract: please provide information on the methodology section.

Authors’ reply: Corrected according to suggestions

2.- Methods:

2.1. Data sources and search strategy: Please provide a table with the number of records located in each database and the final records included in each database. In addition, it is very interesting to include the search date.

Authors’ reply: Corrected according to suggestions, lines 97-112 and the table S1 (Table S1, Supplementary material: The study selection process and search strategy) was added 

2.2. Selection of studies: The review included only the full text (original articles or concise communications) in English. The Cochrane handbook (2011) says "2.3.2: “So far as is possible, it is important to take an international perspective. The evidence collected should not be restricted by nationality or language without good reason, background information such as prevalence and morbidity should where possible take a global view, and some attempt should be made to put the results of the review in a broad context”. The authors should explain why they selected articles in English only. In addition, a short paragraph could be included in the limitations section.

Authors’ reply: Corrected according to suggestions, the "Study limitations" was added, lines 525-531 and a short explanatory paragraph has been added to the text, lines 89-91 and 110-112.

Reviewer 2 Report

The manuscript is overall much improved from the initial submission, with a more robust search of the literature and a thorough and detailed evaluation of the available data.

It seems that there is variability among studies with regards to the definitions of EOS and LOS, and the authors illustrate this variability. Though some of this variability may contribute to the causal organism, it does not seem to be the sole cause of the discrepencies in the organisms, and the authors provide explanation.

In all, the manuscript provides a detailed illustration of the entities of EOS and LOS, and a through summary of the potential interventions and clinical implications of the conditions.

Author Response

Thank you for this comment!